# Online Inertial Machine Learning for Sensor Array Long-Term Drift Compensation

**Xiaorui Dong** [1,2,*] **, Shijing Han** [1,3] **, Ancheng Wang** [1] **and Kai Shang** [2]

1 Institute of Geospatial Information, Information Engineering University, Zhengzhou 450001, China; hsjmsf@163.com (S.H.); acwang_xd@163.com (A.W.)
2 Key Laboratory of Intelligent Information Processing, Shengli College, China University of Petroleum, Dongying 257000, China; 2017021@slcupc.edu.cn
3 School of Natural Resources and Surveying, Nanning Normal University, Nanning 530001, China
* Correspondence: dongxiaorui@slcupc.edu.cn

**Abstract:** The sensor drift problem is objective and inevitable, and drift compensation has essential research significance. For long-term drift, we propose a data preprocessing method, which is different from conventional research methods, and a machine learning framework that supports online self-training and data analysis without additional sensor production costs. The data preprocessing method proposed can effectively solve the problems of sign error, decimal point error, and outliers in data samples. The framework, which we call inertial machine learning, takes advantage of the recent inertia of high classification accuracy to extend the reliability of sensors. We establish a reasonable memory and forgetting mechanism for the framework, and the choice of base classifier is not limited. In this paper, we use a support vector machine as the base classifier and use the gas sensor array drift dataset in the UCI machine learning repository for experiments. By analyzing the experimental results, the classification accuracy is greatly improved, the effective time of the sensor array is extended by 4–10 months, and the time of single response and model adjustment is less than 300 ms, which is well in line with the actual application scenarios. The research ideas and results in this paper have a certain reference value for the research in related fields.

**Keywords:** long-term drift compensation; inertial machine learning; online self-training; data preprocessing; support vector machine; gas sensor array; machine olfactory; chemical sensing

## 1. Introduction

Machine olfactory technology [1] plays an important role in food safety [2], medical care [3], environmental monitoring [4], aerospace [5], and other fields, and this importance is becoming more and more obvious with the development of economy and society. A machine olfactory system typically consists of an array of gas sensors that chemically react with the detected gas to collect sensing data and machine learning methods that analyze and process the data [6].

The gas sensor is composed of chemically sensitive materials connected to the sensor, and the measurement task is accomplished by allowing the molecules of the substance being analyzed to interact with the chemically sensitive materials of the sensor [7]. Gas sensor array is an important part of contemporary Internet of Things (IoT) technology, and its market scale maintains a rapid growth. According to Yole Développement's survey, gas sensors are expected to be worth $2 B in 2026, up from $1.1 B in 2020, with a compound annual growth rate of 10.9% [8]. In recent years, with the development of IoT technology, gas sensing technology has been closely combined with intelligent industry and has been widely studied and applied in food detection, animal and plant breeding, air detection, disease diagnosis, industrial site, pipeline leak detection, hazard monitoring, and other fields [6,7,9–11]. The mechanism of gas sensor detection is that the gas to be analyzed chemically reacts on the surface of the sensor, causing a potential difference inside the

sensor, and then the gas information is converted into an electrical signal to indicate the composition or concentration of the gas. Compared with the traditional gas detection methods, the gas sensor detection method has the advantages of low cost, quick response, ease of use, and great potential and market in the field of gas detection [10]. Ideally, the gas sensor array will always have the same response value when in the same gas or gas mixture and will immediately return its baseline initial value when detection is stopped. However, this ideal situation cannot be achieved in real applications. After a gas sensor is used for a period, it will drift due to factors such as hardware aging and external pollutant poisoning. Drift means that there will be inconsistencies in the sensor response results when detecting the same gas in the same environment. The drift phenomenon can cause the gas recognition model to fail in a relatively short period of time (a few weeks or months). Sensor drift can interfere with gas classification and gas concentration prediction, which has always been the most serious problem faced by gas sensors [1,12].

Sensor drift is objective and inevitable [13]. When a sensor drifts, its performance will decrease, resulting in the inaccuracy of the collected data. Many scholars have performed research on drift compensation from hardware or software perspectives [14]. Due to the limitation of hardware technology, there is no stable, reliable, and inexpensive self-compensating sensor, which is very difficult and costly to develop [10,15]. Compared with hardware compensation, software compensation is cheaper and easier to implement, mainly including univariate methods, multivariate methods and artificial intelligence methods [14]. Both univariate and multivariate methods seek to correct the sensor signals, while the artificial intelligence methods assume the classification model will grasp and learn the drift features on its own. In univariate methods, each sensor calibrates its signal without considering other sensors, making this type of method simpler to implement and widely used. To ensure a good accuracy, a high sampling frequency is needed to detect drift and compensate it in time [14,16,17]. In a sensor array, the moment of drift phenomenon is different for different sensors. Using this law, the multivariate methods integrate and analyze the signals of different sensors in the array to quickly identify the drift phenomena and correct the signals of drifted sensors according to the signals of non-drifted sensors [14,15]. Compared with the univariate methods, the algorithms of the multivariate methods are more difficult to design. Artificial intelligence methods mainly include statistical machine learning methods [1,12,18] and deep learning methods [14,19], the former mainly include support vector machines, multilayer perceptual machines, random forests, etc., and the latter mainly include convolutional neural networks and recurrent neural networks, etc.

At present, many scholars have conducted in-depth research on drift compensation. There is less research on hardware, mainly Sasago [20] created a FET-type hydrogen sensor with a fast response time and low drift. Most of the related research is carried out from the perspective of software compensation. Vergara [15] contributed the famous gas sensor array drift dataset and proposed an integrated learning method based on support vector machines, which provided a machine learning solution for drift compensation research. For more convenient and practicable procedures, Zhu [21] presented a calibration model for classification based on a single category of drift correction samples. Ma [22] developed an online drift compensation model by adapting two domain adaptation-based strategies for online learning. Liu [23] developed a novel active learning methodology that intelligently selects sample labels for drift correction to tackle the issues of only a few drifted samples being usable for label querying. Jiang [24] proposed a unique drift compensation approach based on balanced distribution adaptation, which uses the weight balance factor to adjust the conditional and marginal distributions between the two different domains. In recent years, scholars have started to focus on deep learning techniques and introduced them into the study of drift compensation. Zhao [14] tried to use long short-term memory (LSTM) neural networks to improve the drift compensation effect, and Feng [19] proposed a very innovative method called augmented convolutional neural network (ACNN), which converts sensor signals into matrices and hands them over to convolutional neural network

(CNN) for processing like pictures. Most of the extant studies tend to compensate or analyze sensor data in the form of strong rules or fixed parameter models, which are excessively dependent on historical data and experience and have a large degree of subjectivity. Almost all current studies about online data preprocessing and analyzing are aimed at short-term drift compensation, that is, batch $k$ is used as the sample to predict batch $k$ itself or batch $k + 1$. The studies on long-term drift compensation are mostly based on post-event data processing, which does not have good real-time performance. Deep learning-based approaches, in either training or application sessions, have high hardware requirements and can significantly increase the production cost of sensors. Overall, the existing studies have improved the service life of the sensor to a certain extent, but it is far from enough. If long-term drift compensation can be realized, the sensor life can be extended more effectively, but there are few relevant studies.

Data processing is an indispensable part of machine olfactory system, which directly determines the output of machine olfactory system. We try to study drift compensation from the perspective of improved data processing algorithm. Inspired by inertial navigation technology [25], we propose a novel method (named inertial machine learning) that can help machine learning classifiers realize drift compensation. The inertial navigation technology measures the angular velocity and acceleration information of the carrier relative to the inertial space through inertial measurement components and uses Newton's law of motion to automatically calculate the carrier's instantaneous velocity and position information. Inertial navigation is a completely autonomous navigation technology, and its error (also called drift) increases with navigation time [26]. The drift phenomenon of chemical sensors is very similar to the error accumulation of inertial navigation. Since drift is a gradual process, the record at time $t$ is closer to the true value than the record at time $t + n$, and the accuracy rate of batch $k$ is more likely to be higher than that of batch $k + 1$. Like the improved principle of inertial navigation, our method tries to use as accurate data as possible to train the classifier, making use of the inertia with high accuracy classification effect in a short term to maintain high accuracy for a longer time to achieve long-term drift compensation to some extent. The inertial learning model proposed in this paper uses several queues with upper capacity to store data, so that the model has the characteristics of memory and forgetting. The model uses support vector machine (SVM) as the basic classifier and is validated experimentally with gas sensor array drift data set, which can be obtained online from UCI machine learning repository. Experimental results show that this model can prolong the effective drift compensation time and extend the reliable service time of the sensor by 4–10 months.

The rest of this article is organized, as follows. Section 2 is data preprocessing. Section 3 describes the inertial machine learning method. Section 4 consists of the experimental details and comparative analysis, and finally, conclusions will be drawn in Section 5.

## 2. Data Preprocessing

The sensor array long-term drift dataset is large for two main reasons. The first reason is that the dimension of data collected by sensor array is usually relatively large. The second reason is that to study long-term drift, the period of data acquisition is required to be particularly long. The question of how to properly compress data or extract key information under limited hardware constraints is very important for online model training and data analysis.

### 2.1. Data Acquisition

The gas sensor array drift dataset (GSAD), which is one of the famous data sets of gas sensor drift problems, was adopted as the research object in this study. The dataset, which was created and donated by Alexander Vergara [15] in 2012, contains 13,910 chemical gas sensor data collected by 16 chemical gas sensors (including four TGS2600, four TGS2602, four TGS2610, and four TGS2620) for six different concentrations of different gases (including ethanol, ethylene, ammonia, acetaldehyde, acetone, and toluene). The

data was collected during a 36-month period from January 2008 to February 2011 at the Gas Delivery Platform facility of the Chemical Signals Laboratory at the BioCircuits Institute at the University of California, San Diego. The dataset is divided into 10 batches by time, as shown in Table 1.

**Table 1.** Data distribution of each batch in the GSAD dataset.

| Batch Id | Month Ids | Quantity and Proportion of Each Gas in the Batch | | | | | | | | | | | |
|---|---|---|---|---|---|---|---|---|---|---|---|---|---|
| | | Ethanol | | Ethylene | | Ammonia | | Acetaldehyde | | Acetone | | Toluene | |
| batch1 | 1, 2 | 90 | 20.2% | 98 | 22.0% | 83 | 18.7% | 30 | 6.7% | 70 | 15.7% | 74 | 16.6% |
| batch2 | 3, 4, 8, 9, 10 | 164 | 13.2% | 334 | 26.8% | 100 | 8.0% | 109 | 8.8% | 532 | 42.8% | 5 | 0.4% |
| bacth3 | 11, 12, 13 | 365 | 23.0% | 490 | 30.9% | 216 | 13.6% | 240 | 15.1% | 275 | 17.3% | 0 | 0.0% |
| batch4 | 14, 15 | 64 | 39.8% | 43 | 26.7% | 12 | 7.5% | 30 | 18.6% | 12 | 7.5% | 0 | 0.0% |
| batch5 | 16 | 28 | 14.2% | 40 | 20.3% | 20 | 10.2% | 46 | 23.4% | 63 | 32.0% | 0 | 0.0% |
| batch6 | 17, 18, 19, 20 | 514 | 22.3% | 574 | 25.0% | 110 | 4.8% | 29 | 1.3% | 606 | 26.3% | 467 | 20.3% |
| batch7 | 21 | 649 | 18.0% | 662 | 18.3% | 360 | 10.0% | 744 | 20.6% | 630 | 17.4% | 568 | 15.7% |
| batch8 | 22, 23 | 30 | 10.2% | 30 | 10.2% | 40 | 13.6% | 33 | 11.2% | 143 | 48.6% | 18 | 6.1% |
| batch9 | 24, 30 | 61 | 13.0% | 55 | 11.7% | 100 | 21.3% | 75 | 16.0% | 78 | 16.6% | 101 | 21.5% |
| batch10 | 36 | 600 | 16.7% | 600 | 16.7% | 600 | 16.7% | 600 | 16.7% | 600 | 16.7% | 600 | 16.7% |

### 2.2. Feature Extraction

The dataset contains 128 feature vectors, as shown in Table 2. Vergara et al. focused at two different types of characteristics that leverage the entire dynamic process that occurs at the sensor surface, such as those that reflect the sensor element's adsorption, desorption, and steady-state response [14,15]. $S_i$ in Table 2 represents the $i$th sensor, and each sensor has eight feature vectors, which are $\Delta R$, $\|\Delta R\|$, $ema_{0.001}I$, $ema_{0.01}I$, $ema_{0.1}I$, $ema_{0.001}D$, $ema_{0.01}D$, and $ema_{0.1}D$, respectively [14]. $\Delta R$ represents the difference of the maximal resistance change and the baseline; $\|\Delta R\|$ represents the ratio of the maximal resistance and the baseline values; $ema_\alpha$ means the exponential moving average that converts the increasing/decaying and saturating discrete time series collected from the chemical sensor into a real scalar; $\alpha$ is the scalar being a smoothing parameter of the operator that defines both the quality of the feature and the time of its occurrence along the time series, and it has three different values: 0.1, 0.01, and 0.001; $I$ and $D$ represent the rising transient portion and decaying transient portion of sensor response respectively.

**Table 2.** Placement order of extracted features in the feature vector.

| Features (S1) | Features (S2) | Features (S3) | . . . | Features (S16) |
|---|---|---|---|---|
| 1. $\Delta R\_S1$ | 9. $\Delta R\_S2$ | 17. $\Delta R\_S3$ | . . . | 121. $\Delta R\_S16$ |
| 2. $\|\Delta R\|\_S1$ | 10. $\|\Delta R\|\_S2$ | 18. $\|\Delta R\|\_S3$ | . . . | 122. $\|\Delta R\|\_S16$ |
| 3. $ema_{0.001}I\_S1$ | 11. $ema_{0.001}I\_S2$ | 19. $ema_{0.001}I\_S3$ | . . . | 123. $ema_{0.001}I\_S16$ |
| 4. $ema_{0.01}I\_S1$ | 12. $ema_{0.01}I\_S2$ | 20. $ema_{0.01}I\_S3$ | . . . | 124. $ema_{0.01}I\_S16$ |
| 5. $ema_{0.1}I\_S1$ | 13. $ema_{0.1}I\_S2$ | 21. $ema_{0.1}I\_S3$ | . . . | 125. $ema_{0.1}I\_S16$ |
| 6. $ema_{0.001}D\_S1$ | 14. $ema_{0.001}D\_S2$ | 22. $ema_{0.001}D\_S3$ | . . . | 126. $ema_{0.001}D\_S16$ |
| 7. $ema_{0.01}D\_S1$ | 15. $ema_{0.01}D\_S2$ | 23. $ema_{0.01}D\_S3$ | . . . | 127. $ema_{0.01}D\_S16$ |
| 8. $ema_{0.1}D\_S1$ | 16. $ema_{0.1}D\_S2$ | 24. $ema_{0.1}D\_S3$ | . . . | 128. $ema_{0.1}D\_S16$ |

### 2.3. Data Cleaning and Normalization

Although there are no null data in the data set studied, the possibility of their occurrence should also be considered, so we add a null data check and index reconstruction to the algorithm design to enhance the universality and robustness of the data processing. We hope that the algorithm studied can carry out online adaptive learning and data analysis to adapt to real application scenarios. Therefore, it is impossible to obtain the full view of the data domain and accurate feature correlation in the field of relatively small sample training sets. Due to this consideration, we use min–max normalization to process the data set, and the deeper analysis of the reason for this is explained in Section 2.4. All data items are processed with absolute value as shown in Formula (1), and the processed data items mainly include $ema_\alpha D$ and some of the anomalies that are most likely due to errors in the data collation process. For the outliers suspected to be abnormal in the data set, we used two methods. Method 1 was shown in Formula (2) to correct the problem of decimal point dislocation, and Method 2 set an upper threshold $\theta$ to compress outliers to a moderate range, as shown in Formula (3). The data normalization method adopts min–max normalization, as shown in Formula (4).

$$X = |X|, \tag{1}$$

$$x_{i,j} = \begin{cases} x_{i,j}/10, \, x_{i,j}/X.mean_j \in [5, 20] \\ x_{i,j}/100, \, x_{i,j}/X.mean_j \in [50, 200] \end{cases}, \tag{2}$$

$$x_{i,j} = \min(x_{i,j}, \, X.mean_j/\theta), \tag{3}$$

$$x_{i,j} = x_{i,j}/X.max_j, \tag{4}$$

where $X$ is the sample feature vectors of the data set, $x_{i,j}$ is the entry in the $i$th row and the $j$th column of $X$, $X.mean_j$ is the mean value along the $j$th column direction, $\theta$ is the upper threshold of the compression outliers and its recommended value is 0.2, and $X.max_j$ is the max value along the $j$th column direction. The process of data cleaning and normalization is shown in Figure 1.

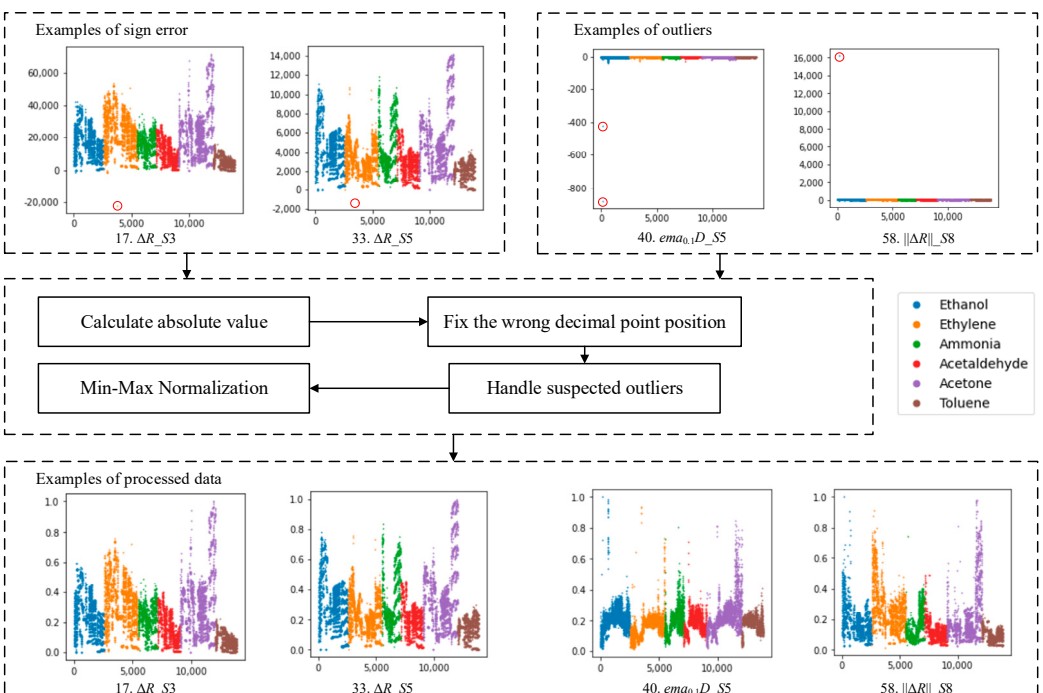

**Figure 1.** The process of data cleaning and normalization.

*2.4. Additional Notes*

At present, principal component analysis (PCA) or the Pearson product moment correlation coefficient (PPMCC) is used in most relevant studies to analyze or process the whole data set. PCA's main goal is to reduce the dimensionality of a data set with a lot of interconnected variables while keeping as much variance as feasible [27]. PPMCC is a measure of the correlation of two variables X and Y measured on the same object or organism, that is, a measure of the tendency of the variables to increase or decrease simultaneously [28]. Many scholars use the two algorithms to reduce the dimensionality of the data set, and the classification effect of the data set processed by these algorithms is much better than that of the unprocessed case. However, we believe that it is impossible to know the full range of data in the vast majority of scenarios, and what we have mastered should be a relatively small number of training set samples. On the other hand, different sensors, different operating environments, different gases to be examined, and different frequencies of use cause different degrees of drift. If the processed data set is divided into the training set and the test set to prove the effect of an algorithm, it means that the algorithm or model is artificially helped to know the distribution of data in advance, thereby improving the experimental effect. Although the two methods have a good effect, they cannot be applied in practical application scenarios, especially when the training set is small. Therefore, we prefer to conduct experiments without correlation analysis. In addition, the Z-Score method requires accurate expectation and variance, but both will change significantly when the sensor drifts. Based on the above consideration, we choose a simplified min–max normalization as shown in Formula (4) to rescale the data set. In this min–max normalization method, the lower bound is zero and the upper bound is the largest value in each dimension of the training set. Even in the test set (or the actual application environment), if there is a signal larger than the set upper bound, it can be reduced to a smaller value under the action of Formula (3) to avoid becoming an outlier and affecting the entire sample set.

## 3. Inertial Machine Learning Method

The study of Vagrin [15] proved that the machine learning algorithm's effect continues to decrease as time goes on. We want to develop algorithms that can take advantage of the inertia of high accuracy in the short term to continue or extend the validity of the algorithm. In addition, considering the need for online data processing, this algorithm cannot have high hardware requirements.

*3.1. Online Inertial Learning Framework*

In this paper, an online inertial machine learning framework with memory and forgetting abilities is proposed, which dynamically adjusts training samples and generates new classifiers for training. Based on the training of sample set at time $t$, the classifier at time $t$ is obtained, which classifies the data to be detected at the next time $t + 1$. After the classification results are processed and integrated, the sample set at time $t + 1$ is generated, and this cycle is followed. The purpose of the forgetting mechanism is to ensure that data sets do not grow large over time and to reduce the undue influence of long-term historical data. The memory mechanism of the framework is also designed to keep the data set balanced as much as possible to ensure that it fits as many base classifiers as possible. The existing drift compensation models can be incorporated into the framework to further improve the recognition effect. The operating mechanism and workflow of the framework are shown in Figure 2.

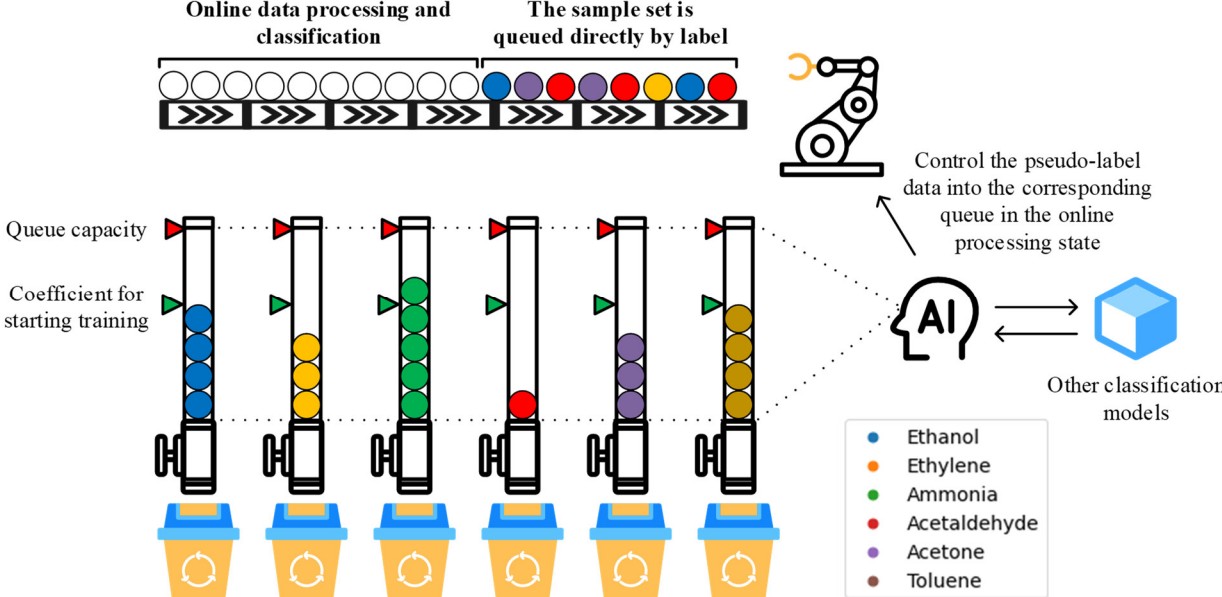

**Figure 2.** The operating mechanism and workflow of the online inertial learning framework.

In the framework, the queue data structure is chosen as the vehicle to remember historical data. The number of queues is the same as the number of types *k* of gases to be detected. All the data in the *k* queues form the training set, which is used to train the classifier. The data in the queue follows a first-in, first-out principle, while the data in the training set is unordered. In the initial phase, the sample is entered into a corresponding queue $Q_i$ according to its label *i*. When the number of data items in *k* queues is greater than the set coefficient $\ell$ for starting training, the model starts training and generates the classifier based on the sample set at the time. In this phase, after the classifier completes the data preprocessing at time *t*, the pseudo-label of the data at time *t* is obtained, and the pseudo-label data is regarded as the real sample and entered the corresponding queue, and so on. When the number of data items in a queue $Q_i$ exceeds the rated capacity, the data items in the queue $Q_i$ are removed from the queue so that the data in each queue do not grow indefinitely over time. The framework is designed to take advantage of the inertia of short-term classification with high accuracy to extend the effective classification time.

*3.2. Description of Each Phase and Algorithm Design*

From the perspective of storage queues, there are three main phases as follows:

1.  Initially, the storage queues are empty. Data items are queued in time series to build the initial sample set. In real applications, this process is the data initialization phase, which can be done in the lab or using calibration data to populate the queue. The end milestone of this phase is that the number of data items in all queues reaches the start learning coefficient.

2.  In this phase, the starting learning conditions are reached, the classifier starts to be trained, and no real sample data will be queued. The pseudo-label data enter the queues sequentially as real samples. If the number of data items in a queue reaches the upper limit, every time the pseudo-label data enters the queue, the queue of the corresponding category will perform the dequeue operation accordingly. The end milestone of this phase is that the number of data items in all queues reaches the upper limit (i.e., queue capacity).

3.  All storage queues in this phase are full. The classifier will continue to work and continuously enqueue the pseudo-label data predicted by the classifier, and each enqueue is accompanied by a dequeue operation.

The coefficient for starting learning is defined as $\ell$, and the queue capacity as $\lambda$, which can be expressed as follows:

$$\begin{aligned} \ell &= (\ell_1, \ell_2, \ell_3, \cdots, \ell_k) \\ \lambda &= (\lambda_1, \lambda_2, \lambda_3, \cdots, \lambda_k) \end{aligned} \tag{5}$$

These two coefficients need to be set manually. The principle of setting the starting learning coefficient $\ell$ is based on the sample set, and its value is usually equal to the amount of data in the sample set. If the data set is extremely unbalanced, in order to ensure the stability of the algorithm, it is necessary to discard some redundant data and retain the latest sample data. The setting of the queue capacity $\lambda$ not only limits the unlimited growth of data but also controls the balance of training samples as much as possible. The settings of the two coefficients should follow:

$$\forall i \in [1, k], \ \lambda_i \geq \ell_i, \tag{6}$$

Otherwise, it will not be able to enter the online learning and data analysis phase.

### 3.3. Evaluation Method

The accuracy rate (ACC) is the main algorithm evaluation index of this study, and its formula is shown as follows:

$$ACC = (TP + TN)/(TP + FP + FN + TN), \tag{7}$$

where $TP$ (true positives) means that the actual class is positive and the predicted class is also positive, $FP$ (false positives) means that the predicted class is positive but actual class is negative, $FN$ (false negatives) means that the predicted class is negative but actual class is positive, and $TN$ (true negatives) means that the actual class is negative and the predicted class is also negative.

### 3.4. Base Classifier

The classification algorithms or models in this framework are not limited. The support vector machine (SVM) [29] is a very popular choice in most relevant researches. The SVM has strict mathematical theory support and strong generalization ability, has high classification effect in dealing with high-dimensional small sample problems, and has good effect in short-term drift compensation [1,10,14,15]. In this study, the support vector machine was used as the base classifier, the radial basis function was used as the kernel function, the penalty parameter was set to 1, and the one-to-one method was adopted to achieve multi-classification. Only the support vector machine is used to classify the data set, and the classification results are sorted and analyzed. Tables 3 and 4 show the classification effects of different training sets and test sets.

**Table 3.** The classification accuracy of only SVM (use batch m as the training set and batch n as the testing set where m = 1, 2, . . . , 9 and n = m + 1, m + 2, . . . , 10).

| Train Set Batch | ACC (%) of Test Set Batch | | | | | | | | |
|---|---|---|---|---|---|---|---|---|---|
| | 2 | 3 | 4 | 5 | 6 | 7 | 8 | 9 | 10 |
| 1 | 76.21 | 49.43 | 33.54 | 23.85 | 33.73 | 33.29 | 25.51 | 34.25 | 41.41 |
| 2 | | 90.16 | 86.95 | 68.02 | 42.04 | 42.56 | 31.29 | 59.36 | 37.47 |
| 3 | | | 69.56 | 94.92 | 72.17 | 73.45 | 40.81 | 61.7 | 49.66 |
| 4 | | | | 86.29 | 45.56 | 39.8 | 17.68 | 22.97 | 14.77 |
| 5 | | | | | 56.43 | 44.45 | 39.79 | 43.61 | 19.27 |
| 6 | | | | | | 78.24 | 75.17 | 36.8 | 51.77 |
| 7 | | | | | | | 86.05 | 65.31 | 62.61 |
| 8 | | | | | | | | 61.27 | 20.02 |
| 9 | | | | | | | | | 25.05 |

**Table 4.** The classification accuracy of only SVM (use batch 1-m as the training set and batch n as the testing set where m = 1, 2, . . . , 9 and n = m + 1, m + 2, . . . , 10).

| Train Set Batch | ACC (%) of Test Set Batch | | | | | | | | |
|---|---|---|---|---|---|---|---|---|---|
| | 2 | 3 | 4 | 5 | 6 | 7 | 8 | 9 | 10 |
| 1 | 76.21 | 49.43 | 33.54 | 23.85 | 33.73 | 33.29 | 25.51 | 34.25 | 41.41 |
| 1–2 | | 88.27 | 86.95 | 87.3 | 32.52 | 43.75 | 29.25 | 53.4 | 38.91 |
| 1–3 | | | 87.57 | 95.43 | 69.69 | 68.44 | 55.1 | 74.25 | 43.08 |
| 1–4 | | | | 96.95 | 69.6 | 66.95 | 52.72 | 72.34 | 42.58 |
| 1–5 | | | | | 72.39 | 72.57 | 54.08 | 72.97 | 43.91 |
| 1–6 | | | | | | 85.8 | 90.13 | 67.44 | 54.3 |
| 1–7 | | | | | | | 90.81 | 76.38 | 65.19 |
| 1–8 | | | | | | | | 77.02 | 67.75 |
| 1–9 | | | | | | | | | 66.77 |

## 4. Experiments and Results

### 4.1. Experimental Datasets and Environment

We designed data sets dataset1 and dataset2 for long-term drift study according to the following rules.

- dataset1: Use batch 1 as the training set and batch k as the test set, where k = 2, 3, 4, 5, 6, 7, 8, 9, 10.
- dataset2: Use batch 1–2 as the training set and batch k as the test set, where k = 3, 4, 5, 6, 7, 8, 9, 10.

Since there is no toluene gas sample in batch 3–5, it is meaningless to use batch 1–3, batch 1–4 or batch 1–5 as the training set. In addition, the assumption of using this framework is that the samples are collected in the laboratory or calibrated manually, so using longer-term data as the training sets is not in line with practical application scenarios.

This study used Anaconda (Python 3.8, individual edition) as the development environment. The experimental environment used Intel i5-6200U 2.40 GHz CPU, 8 GB RAM. This project required third-party libraries including NumPy 1.20.1, Pandas 1.2.4, SciPy 1.6.2, Scikit-learn 0.24.1, and Matplotlib 3.3.4.

### 4.2. Experimental Results for Dataset1

Set $\ell = (90, 98, 83, 30, 70, 74)$ according to the data distribution of batch1. To keep the data set balanced, let the values in $\lambda$ be the same. Define $\lambda_i = c$ where $i = 1, 2, \cdots, 6$ and $c$ is a constant. Set $c$ equal to 100, 200, 300, 400, 500, and 600, respectively, and conduct the experiment. Experimental results, total time (TT), and average step time (AST) are shown in Table 5.

**Table 5.** Experimental results and average time for dataset1.

| $c$ | ACC (%) of Test Set Batch | | | | | | | | | TT (s) | AST (ms) |
|---|---|---|---|---|---|---|---|---|---|---|---|
| | 2 | 3 | 4 | 5 | 6 | 7 | 8 | 9 | 10 | | |
| 100 | 68.09 | 47.29 | 38.51 | 33.50 | 46.96 | 19.07 | 19.39 | 11.91 | 37.72 | 231.85 | 17.22 |
| 200 | 75.08 | 76.23 | 45.96 | 63.45 | 69.74 | 22.53 | 11.22 | 0.0 | 22.86 | 431.94 | 32.08 |
| 300 | 87.94 | 83.35 | 45.96 | 64.97 | 56.91 | 29.69 | 39.12 | 11.70 | 23.39 | 622.45 | 46.23 |
| 400 | 87.94 | 83.35 | 80.74 | 73.10 | 57.09 | 46.50 | 33.00 | 23.62 | 17.22 | 843.44 | 62.64 |
| 500 | 87.94 | 83.35 | 80.75 | 73.10 | 70.83 | 56.88 | 43.20 | 45.11 | 8.31 | 988.20 | 73.39 |
| 600 | 87.94 | 83.42 | 62.73 | 73.10 | 70.83 | 60.92 | 43.54 | 45.74 | 10.64 | 1232.71 | 91.55 |

### 4.3. Experimental Results for Dataset2

Set $\ell = (254, 432, 183, 139, 602, 79)$ according to the data distribution of batch1–2. Set $c$ equal to 602, 700, 800, 900, 1000, 1100, and 1200, respectively, and conduct the experiment. Experimental results and average time are shown in Table 6.

**Table 6.** Experimental results and average time for dataset2.

| c | ACC (%) of Test Set Batch | | | | | | | | TT (s) | AST (ms) |
|---|---|---|---|---|---|---|---|---|---|---|
| | 3 | 4 | 5 | 6 | 7 | 8 | 9 | 10 | | |
| 602 | 98.80 | 91.93 | 96.45 | 69.87 | 60.20 | 37.07 | 38.09 | 23.22 | 1584.59 | 129.66 |
| 700 | 98.80 | 90.68 | 96.45 | 72.04 | 65.65 | 46.26 | 51.28 | 26.81 | 1804.33 | 147.64 |
| 800 | 98.80 | 90.68 | 96.45 | 72.04 | 65.57 | 46.94 | 50.85 | 25.94 | 2040.36 | 166.96 |
| 900 | 98.80 | 90.68 | 95.43 | 72.04 | 59.56 | 42.86 | 45.74 | 32.14 | 2444.38 | 200.01 |
| 1000 | 98.80 | 90.68 | 95.43 | 72.04 | 59.75 | 43.54 | 46.17 | 23.86 | 2809.77 | 229.91 |
| 1100 | 98.80 | 90.68 | 95.43 | 72.22 | 67.37 | 47.96 | 51.28 | 30.67 | 3269.14 | 267.50 |

### 4.4. Experimental Comparison and Analysis

Through the analysis of the above experimental results, the experimental effect of dataset2 is higher than that of dataset1, indicating that sufficient initial samples significantly improve the results. The higher the value of $c$ is, the stronger the final classification ability will be. At the same time, the increase of $c$ will also lead to longer storage queues, more complex calculation, and ultimately more time consumption. When $c$ reaches about 2–5 times the $\max(\ell)$, the growth efficiency of accuracy begins to slow down. Therefore, we can try to set $c$ to be equal to $\max(\ell) * b$ where $b \in [2, 5]$. Currently, due to the lack of more research data, it is impossible to conduct in-depth research on the value of $\lambda$. In practical applications, the value of $\lambda$ should also be determined based on the data processing capacity and real-time requirements of sensors. In dataset1, the relatively good case is $c = 500$, while in dataset2, the relatively good case is $c = 1100$. The experimental comparison between ours and only SVM is shown in Tables 7 and 8, and Figure 3. The comparison of the computing times is shown in Figure 4.

**Table 7.** Experimental comparison for dataset1.

| | ACC (%) of Test Set Batch | | | | | | | | |
|---|---|---|---|---|---|---|---|---|---|
| | 2 | 3 | 4 | 5 | 6 | 7 | 8 | 9 | 10 |
| Only SVM | 76.21 | 49.43 | 33.54 | 23.85 | 33.73 | 33.29 | 25.51 | 34.25 | 41.41 |
| Ours ($c = 500$) | 87.94 | 83.35 | 80.75 | 73.1 | 70.83 | 56.88 | 43.2 | 45.11 | 8.31 |
| improvement value | 11.73 | 33.92 | 47.21 | 49.25 | 37.1 | 23.59 | 17.69 | 10.86 | −33.1 |
| improvement ratio (%) | 15.39 | 68.62 | 140.76 | 206.5 | 109.99 | 70.86 | 69.35 | 31.71 | −79.93 |

**Table 8.** Experimental comparison for dataset2.

| c | ACC (%) of Test Set Batch | | | | | | | |
|---|---|---|---|---|---|---|---|---|
| | 3 | 4 | 5 | 6 | 7 | 8 | 9 | 10 |
| Only SVM | 88.27 | 86.95 | 87.3 | 32.52 | 43.75 | 29.25 | 53.4 | 38.91 |
| Ours ($c = 1100$) | 98.8 | 90.68 | 95.43 | 72.22 | 67.37 | 47.96 | 51.28 | 30.67 |
| improvement value | 10.53 | 3.73 | 8.13 | 39.7 | 23.62 | 18.71 | −2.12 | −8.24 |
| improvement ratio (%) | 11.93 | 4.29 | 9.31 | 122.08 | 53.99 | 63.97 | −3.97 | −21.18 |

Through the analysis of the above experimental results, it is easy to see that the effectiveness of the framework on the base classifier is improved. The decline in classification accuracy is more gradual in both dataset1 and dataset2. Starting with batch 7, their accuracy decreased significantly. Dataset2 also consistently outperforms dataset1 due to the larger training set. For both dataset1 and dataset2, the accuracy and effective time of the online inertial machine learning framework are significantly improved compared to using base classifier only. Using only support vector machines, the performance degrades significantly after about 1–2 batches, while using our framework can extend the effective time by an extra 1–4 batches (about 4–10 months). From the perspective of time consumption, the single step time (including the updating of sample set and classification model) is within

300 ms, which has no significant impact on real-time requirements and is suitable for practical application scenarios.

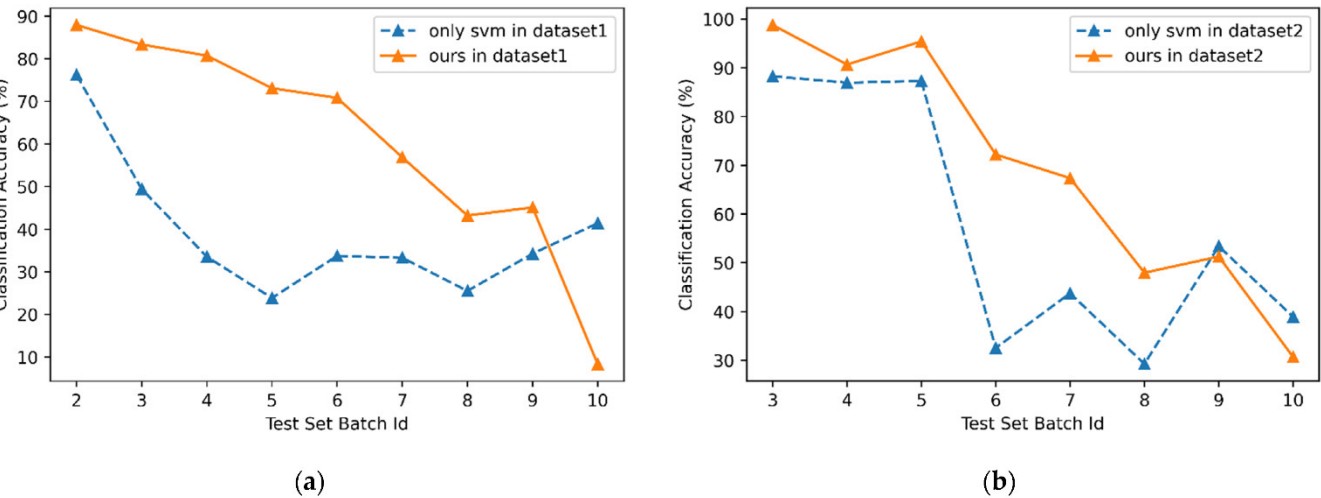

**Figure 3.** The experimental comparison between only svm and ours (**a**) in dataset1 and (**b**) in dataset2.

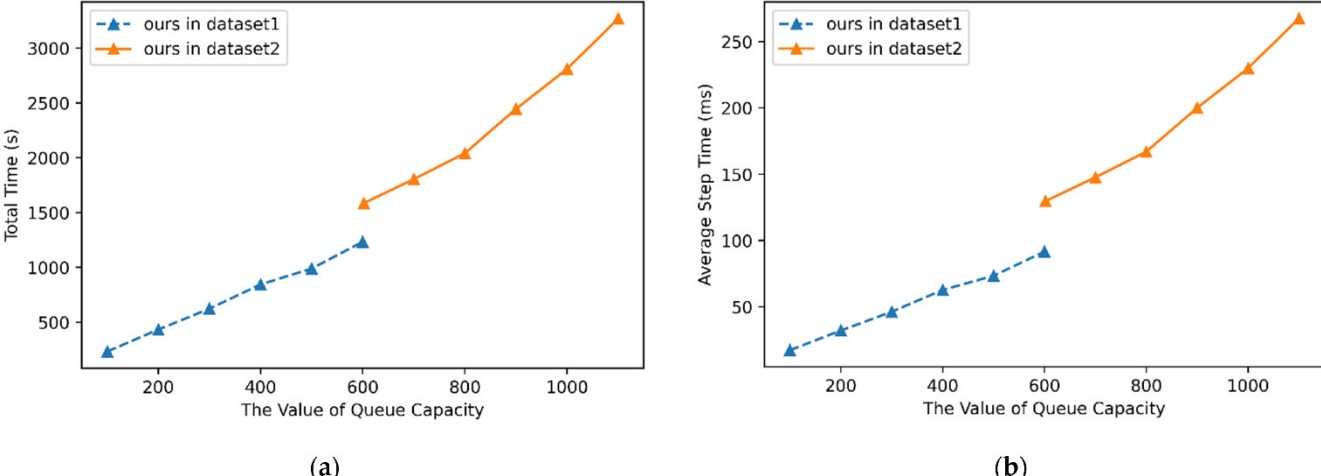

**Figure 4.** The computing time of ours: (**a**) the total time spent processing all batch data; (**b**) the average step time spent processing a record.

## 5. Conclusions

In this paper, we design a data preprocessing method and an inertial machine learning framework that have no special hardware requirements and aim to achieve online long-term drift compensation without increasing manufacturing costs. The data preprocessing method revises a variety of common errors and adopts the min–max normalization, instead of using PCA, PPMCC, and Z-Score methods like most current studies, which can better realize practical engineering application. This method obviously improves the training samples and lays a good foundation for subsequent classification experiments. The inertial machine learning framework takes advantage of the recent inertia of high classification accuracy to extend the reliability of sensors. By analyzing the experimental results on the gas sensor array drift dataset, the classification accuracy is greatly improved, the effective time of the sensor array is extended by 4–10 months, and the time of single response and model adjustment is less than 300 ms, which fits well with realistic application scenarios for low-cost online data processing.

The source code is available at https://github.com/dongxiaorui1988/OIML_SALDC (accessed on 10 December 2021). In the future, more models will be coded and used as the base classifier to further study the framework proposed in this paper.

**Author Contributions:** Conceptualization, X.D. and K.S.; methodology, X.D.; software, X.D. and K.S.; validation, X.D., S.H. and A.W.; formal analysis, X.D.; investigation, X.D.; resources, S.H.; data curation, X.D.; writing—original draft preparation, X.D.; writing—review and editing, X.D.; visualization, X.D.; supervision, S.H.; project administration, X.D.; funding acquisition, A.W. All authors have read and agreed to the published version of the manuscript.

**Funding:** This research was funded by National Natural Science Foundation of China, grant number 61262047.

**Institutional Review Board Statement:** Not applicable.

**Informed Consent Statement:** Not applicable.

**Conflicts of Interest:** The authors declare no conflict of interest.

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
