# Peer review of "Online Inertial Machine Learning for Sensor Array Long-Term Drift Compensation"

_chemosensors, doi:10.3390/chemosensors9120353_

Round 1

Reviewer 1 Report

Dear authors

In this paper you present an algorithm for the classification of gases with electronic nose data from an open data repository.  The algorithm consists on a queue of samples that discard them based on FIFO. In this way you achieve drift compensation. The idea is original enough, drift is always a problem worth studying, and the algorithm could be incorporate easily in many classification algorithms.

But there are a few points that need to be clarified before publication. For this I think the paper needs a mayor revision.

First the English need to be corrected since in some parts language problems really hinders the understanding of the paper, particularly at the introduction and at 235-242.

Second, a comparison of other papers on the same database should be provided. It can frame the results compared to other studies, there are probably some other studies about the drift, because the database seems oriented to study that problem.

Also, some other points need clarification:  

Lines 70, 73, I don’t understand why the univariate methods are sensitive to sampling rate. Also, other claims made are strange. How multi vs univariate are different? I can’t understand this sentence “Multivariate methods tend to compensate the entire sensor and further respond to signal correction.” Maybe could you give some references to example of those methods to see the differences? Statistical machine learning are multivariate too?

Line 105-107: I don’t know inertial computing, inertial machine learning or inertial navigation technology. Could you give some definitions and references to it?

Line 198: I don’t understand “inertial of accuracy”

Finally, other minor issues:

Line 37-39: Ref for this data, is it [3]?

Line 66-69: There is not hardware compensation or have Ref 7 a hardware compensation? Both can’t be true.

Line 90: Ref to Jian work?

Line 80: How ref 11 fits in your classification of drift compensation?

126: Define before using PPMCC, (move from 164

192: Max and Min are based also on the whole database, and would have the same problems as z-normalization or PCA?

Line 270: Reference for SVM drift compensation ability?

Line 300: I think is better to add computing time at some point.

Reviewer 2 Report

In this work, the author proposed an Online Inertial Machine Learning to realize sensor drift compensation. Based on the UCI data set, the effectiveness of the method is verified. This article has a clear research significance, while the method description and data processing process are clear. Therefore, I suggest that the article be published after some revisions:

(1) The content of the literature review is too brief, and the author should increase the references.

(2) For the convenience of readers, the author's main contribution should be written separately. For example, the main contributions are as follows: (1), (2), (3).

(3) The resolution of Figure 3 is low. It is recommended that the author use Origin or Matlab.

(4) When describing the advantages of SVM, the author should combine the data characteristics of the drift data, and cite references.

(5) The conclusion should show some quantitative evaluation indicators to prove the effectiveness of the sensor drift compensation method proposed in this paper.

Reviewer 3 Report

The sensor drift problem should be solved, and your approach seems to be fascinating. The introduction of your manuscript looks very good. However, it was difficult to understand the effectiveness of your strategy, especially for readers who are not familiar with data processing. Could you add figures for the comparison of sensor drift with and without modification using a support vector machine?

Round 2

Reviewer 1 Report

Dear authors

Lines 60-81 are too similar to ref 14. I think that before publication all this paragraph should be totally rewritten. Also, I still don't understand why the univariate methods are sensitive to sampling rate while multivariate are not, you didn't answer it. I also, I don't understand why multivariate requires frequent sampling and recalibration and others do not. Additionally, please review the references because they don’t always support the claims made, for example reference 15 shows drift compensation based on measuring a reference channel, not any "periodic adjustment of the sensor, such as removing any poisoned and aged modules from the surface film of the sensor".

Another small issue that still could be improved is:

Response 3. Yes, your method improves the original results, and it would be good to be able to explicitly compare to it in the paper (like writing we got xx% and in the original paper they got xx% error...). Also, the paper is nearly 10 years old, some more recent papers should be referenced, for example this one but you can search online for any other one that maybe is more comparable to your paper: https://www.sciencedirect.com/science/article/pii/S0925400521015549

Finally thank you for addressing all other issues:

Response 1. Yes, sorry about this imprecision in the description of your paper, I understood that there are k queues.

Response 2. Than you for the efforts correcting English. I think it improved the paper. 

Response 5, 6, 7, 8, 9, 10, 11, 12, 13, and 14. Thank you for addressing or clarifying all these issues.

Warm regards.

Reviewer 3 Report

The manuscript was modified for people who are not familiar with data processing.

Author Response

We sincerely thank you for thoroughly examining our manuscript and providing very helpful comments to guide our revision.

Round 3

Reviewer 1 Report

 Dear authors

Thank you for answering all the comments

Point 1: I understand now your reasoning, univariate requires data with enough sampling rate to detect and correct the drift, while multivariate can use the data of other sensors to detect and correct the drift of other some sensors.

Point 2. Thank you for incorporating the comparison with other methods. About the CNN in 19, for what I understood viewing the paper, the convolution is made along time and along sensor, so for each gas exposition they have a n x t matrix (n for sensors and t for time). Because the sensors are not independent (gas sensors tend to be highly correlated) the data is not totally independent. But I didn’t read the paper in deep detail so I might be wrong. Anyway, if you think the paper should not be referenced please remove it from your paper. The idea was to frame your study with other papers that you think are good ones (I think a quantitative comparison would have been better, but your qualitative comparison is good enough).

Warm regards.